# Effects of Water Temperature and Photoperiod on the Antioxidant Status and Intestinal Microbiota in Larval Spotted Mandarin Fish, *Siniperca scherzeri*, in the Yalu River

**DOI:** 10.3390/biology14101400

**Published:** 2025-10-13

**Authors:** Jun Yang, Xufang Liang, Yan Wang, Na Li, Yanjun Wang, Ke Lu, Tao Tian, Jiao Li, Yuyu Xiong, Meixuan Li, Yicheng Gao

**Affiliations:** 1College of Fisheries, Chinese Perch Research Center, Huazhong Agricultural University, Wuhan 430070, China; yangj@dlou.edu.cn (J.Y.); xufang_liang@hotmail.com (X.L.); luke@mail.hzau.edu.cn (K.L.); 2Engineering Research Center of Green Development for Conventional Aquatic Biological Industry in the Yangtze River Economic Belt, Ministry of Education, Wuhan 430070, China; 3College of Marine Science Technology and Environment, Dalian Ocean University, Dalian 116023, China; lina2023@dlou.edu.cn (N.L.); wangyj@dlou.edu.cn (Y.W.); 4Center for Marine Ranching Engineering Science Research of Liaoning, Dalian Ocean University, Dalian 116023, China; ttbeyond@126.com; 5College of Fisheries and Life Science, Dalian Ocean University, Dalian 116023, China; lijiao@dlou.edu.cn (J.L.); xiongyuyu@dlou.edu.cn (Y.X.); 6School of Life Sciences, Liaoning University, Shenyang 110036, China; kjxhdl@163.com; 7The First Clinical College, China Medical University, Shenyang 110122, China; gaoyicheng2004@126.com

**Keywords:** *Siniperca scherzeri*, aquaculture, antioxidant enzymes, fish health and bacteria

## Abstract

**Simple Summary:**

The spotted mandarin fish, *Siniperca scherzeri*, is a common artificially farmed species in northern China. Due to the destruction of the ecological environment and overfishing, the wild resources of *S. scherzeri* have gradually decreased and can no longer meet the market demand. Therefore, farming mandarin fish can meet greater market demand. This study investigated the effects of light duration and water temperature variation during mandarin fish farming on hepatic antioxidant capacity and intestinal microbiota, aiming to provide data references for its cultivation.

**Abstract:**

Antioxidant capacity and intestinal microbiota play a crucial role in the nutrition, immunity, and overall health status of fish. It is significant to understand the impact of environmental changes on the health of *Siniperca scherzeri*, an important breeding species. Therefore, in order to investigate the effects of photoperiod (8L: 16D, 12L: 12D and 16L: 8D) and water temperature (18 and 28 °C) on the antioxidant capacity and intestinal microorganisms of *S. scherzeri*, liver and intestinal samples from cultured juvenile *S. scherzeri* were collected for related analyses. The photoperiod group of 12L: 12D and the water temperature group of 18 °C presented a higher abundance of Malondialdehyde (MDA) and lower abundances of Superoxide dismutase (SOD), Glutathione peroxidase (GSH-Px), and Catalase (CAT) than other groups (*p* < 0.05). No significant difference in intestinal microbial diversity was found among different groups (*p* > 0.05), except that the ACE index showed significant differences among different temperature groups (*p* < 0.05). Significant differences in the relative abundances of *Proteobacteria* and *Tenericutes* were found among different groups (*p* < 0.05). Modifying the duration of light exposure could effectively mitigate oxidative reactions and optimal high temperatures could suppress oxidation in juvenile *S. scherzeri*. However, these conditions also influenced the feeding behavior of juvenile *S. scherzeri* and the composition of intestinal microbiota and promoted the proliferation of opportunistic bacteria. The study provides the valuable data of the aquatic habitat of *S. scherzeri.*

## 1. Introduction

Environmental factors, such as stocking density, salinity, water temperature, photoperiod, and light, largely affect the growth and development of aquatic organisms [1,2]. Among these, water temperature is an important abiotic environmental variable and affects the adaptation of animals to the environment [3]. When water temperature is within the appropriate range for aquaculture, the food intake, metabolism rate, and growth rate of fish increase [4]. Once water temperature is beyond this range, the food intake, metabolism rate, and growth rate of fish decrease, and fish may even die [5,6]. In the breeding process, an accurate range of suitable water temperature is required for fish to accelerate growth, shorten the breeding period, save costs, and improve survival rate and yield at various stages [7]. In addition, the increase in water temperature causes the concentration of dissolved oxygen in water to decrease and changes the structure and number of phytoplankton and animals and the habitat of fish [8,9].

Similarity to water temperature, photoperiod also affects the growth, feeding, development, and spawning of fish [10]. The optimization of an artificial light regime can accelerate fish growth [11]. The flexibility of artificial lighting allows for various growth environments of a wide range of cultured species [12]. It was reported that light had significant influences on physiological stress in fish cultures, such as faster growth induced by an increased light duration, the enhanced immune system of fish by specific wavelengths of light, and reduced stress levels [13,14,15]. The photoperiod is one of the most critical characteristics of the light environment. An inappropriate photoperiod may lead to the production of reactive oxygen species (ROS) at varying levels, thereby inducing oxidative stress [15,16]. In addition, Gao et al. found that under both blue and white light conditions, the activities of these enzymes at a 16L: 8D photoperiod were significantly lower than those at a 12L: 12D photoperiod, suggesting that extended light exposure increased antioxidant enzyme activity [17].

The sustainable development of the aquaculture industry plays a crucial strategic role in ensuring global animal protein supply and promoting rural economic development. *Siniperca scherzeri* is an important and valuable freshwater benthic carnivorous fish species throughout the inland waters of China, the Korean Peninsula, and Vietnam [18,19]. *S. scherzeri* is one of the most important fish exported to China due to its tender meat, rich nutrition, and zero intermuscular spines [20]. In recent years, due to the destruction of the ecological environment and overfishing, the wild resources of *S. scherzeri* have gradually decreased and can no longer meet the market demand [20]. Compared with *Siniperca chuatsi* and *Siniperca kneri*, *S. scherzeri* has a smaller body and slower growth rate [21]. Therefore, the supply of natural seedstock alone is insufficient to fully meet market demands [22]. However, against the backdrop of the gradual popularization of intensive aquaculture practices, health issues in fish caused by improper regulation of environmental factors have become increasingly prominent. Among the numerous environmental factors, light and temperature are fundamental and highly variable physical parameters that persist throughout the entire aquaculture cycle, profoundly influencing the physiological rhythms, metabolic rates, and stress responses of fish [23,24]. Therefore, in-depth investigation into the mechanisms through which light and temperature affect the physiological health of mandarin fish is a critical scientific issue that must be addressed to achieve healthy and efficient aquaculture.

The oxidative stress of *S. scherzeri* was analyzed by investigating four key antioxidants, and the responses in the livers of *S. scherzeri* to warm and cold temperatures combined with different photoperiods were compared in this study. In addition, the relationships between environmental microbial communities and gut microbiota were investigated with 16S rRNA sequencing technology in order to reveal the relationships between fish diseases and environmental microbes based on different water temperatures and photoperiods.

## 2. Materials and Methods

### 2.1. Experimental Design and Breeding Management

The experimental larvae were artificially bred from mandarin fish in the Yalu River, and fertilized eggs were hatched in the hatching bucket. On Day 1, larval fish were randomly divided into five groups. Each group had three replicates, and every replicate contained 50 larval fish. Larvae were cultured in transparent basins (6 L) with aerated water. During the experiment, the larval fish were fed with the suitable size of zebrafish three times a day. The water was replaced three times a day to remove food residues and feces. The pH value was 7.53 to 7.70. The dissolved oxygen concentration was 8.5 to 8.9 mg/L. Samples were ready for experimentation after 20 days of incubation.

Among the five groups, three groups were set as photoperiod groups and two groups were set as water temperature groups. For the photoperiod groups, three photoperiods (8L: 16D, 12L: 12D and 16L: 8D (L: Light, D: Dark)) were set, with 25 ± 0.5 °C temperature of breeding water. In the two water temperature groups, water temperature was, respectively, set at 18 °C and 28 °C, with natural daylight hours. Heating rods were used to control ambient water temperature.

Three samples of juvenile *S. scherzeri* were randomly selected from each group after 20 days of different treatments. The fish were anesthetized and then euthanized. Samples were dissected under sterile conditions on a biological safety cabinet. The intestinal contents were carefully extruded using a sterilized glass rod. Liver and intestinal contents were aseptically collected and immediately placed into pre-sterilized 1.5 mL EP tubes, with each tube labeled with the corresponding sample ID. All extracts were rapidly frozen and stored at −80 °C for subsequent experiments.

### 2.2. Ethical Approval

Our research was approved by the Animal Ethics Committee of Dalian Ocean University (Permit Number: DLOU2023008, 8 August 2023) and performed in accordance with the relevant institutional and national guidelines, and the manuscript conforms with the ARRIVE Guidelines for Reporting Animal Research.

### 2.3. Analysis of Oxidative Stress Parameters

Livers were rinsed with pre-cooled normal saline, blotted dry with filter paper, and then homogenized in a pre-chilled glass homogenizer. The levels of Malondialdehyde (MDA) (Cat. No. A003-1), superoxide dismutase (SOD) (Cat. No. A001-1), glutathione peroxidase (GSH-Px) (Cat. No. A005), and catalase (CAT) (Cat. No. A007-1) were analyzed by related kits according to corresponding instructions (Nanjing Jiancheng Bioengineering Institute, Nanjing, China).

### 2.4. Extraction and Sequencing of Microbial DNA

Microbial DNA was extracted from the intestinal contents of *S. scherzeri* samples using the method described in the E.Z.N.A.^®^ Bacteria DNA Kit (Omega Bio-Tek, Norcross, GA, USA). The concentration of DNA was measured (Synergy™ HTX Multiscan spectrum, BioTek, Winooski, VT, USA), and its integrity was assessed via agarose gel electrophoresis. Sequencing adapters were ligated to the primer ends, followed by PCR amplification. The resulting PCR products were then purified, quantified, and normalized to construct the sequencing library. The constructed libraries first underwent quality control checks. Libraries that passed the quality control were sequenced on the Illumina NovaSeq 6000 (Illumina, Inc., San Diego, CA, USA). The V1-V3 regions of 16S rRNA gene were amplified. Universal primers were F (5′-GGACTACHVGGGTWTCTAAT-3′) and R (5′-ACTCCTACGGGAGGCAGCA-3′). The sequencing data (ID PRJNA1333706) generated in this study have been deposited in the Sequence Read Archive (SRA) of the National Center for Biotechnology Information (NCBI) (http://www.ncbi.nlm.nih.gov/, accessed on 25 September 2025). Raw image data files obtained from high-throughput sequencing were converted into raw sequenced reads through base calling analysis. The results are stored in FASTQ files, which contain both the sequence information and the corresponding quality scores.

### 2.5. Processing of Sequencing Data

The raw data were first subjected to quality filtering using Trimmomatic (version 0.33), followed by the identification and removal of primer sequences with Cutadapt (version 1.9.1). The paired-end reads were then assembled, and chimeras were removed using USEARCH (version 10) with the UCHIME algorithm (version 8.1), resulting in high-quality sequences for subsequent analysis.

Operational units (OTUs) were clustered with the similarity cutoff of 97% in UPARSE (Version 10.0). Chimeric sequences were identified and removed with UCHIME. The taxonomy of each 16S rRNA gene sequence was analyzed in RDP Classifier (http://rdp.cme.msu.edu/, accessed on 31 March 2022) against the silva (SSU132) 16S rRNA database with a confidence threshold of 70%.

Alpha diversity was analyzed by QIIME2 (http://qiime2.org, accessed on 31 March 2022) [25]. Chao1 [26] and ACE [27] were used to assess the species richness of samples. Shannon–Wiener indices [28] and the Simpson diversity index were used to assess the diversity of different sample groups. The coverage was used to estimate the authenticity of sequenced data [29]. A heatmap was plotted in Heml 1.0 with the top 20 genera in every group of photoperiod and water temperature samples.

### 2.6. Statistical Analyses

All statistical analyses were carried out with SPSS 22.0 and Excel. Data are expressed as means ± S.E.M of triplicates. Prior to conducting one-way ANOVA, all data were subjected to a One Sample *t*-test, as well as normality and homogeneity tests. A 95% confidence interval was adopted as the criterion for significant differences.

## 3. Results

### 3.1. Activities of Antioxidant Enzymes

The activities of the four antioxidant enzymes are shown in Table 1. The activities antioxidant enzymes showed significant differences among different photoperiod groups (*p* < 0.05). Among the three photoperiod groups, the abundance of MDA was the lowest in the group of 8L: 16D and the highest in the group of 12L: 12D. The MDA value exhibited an initial increase followed by a subsequent decrease. Contrary to the abundance of MDA, the abundances of GSH-Px and CAT firstly decreased and then increased. The abundances of GSH-Px and CAT in the group of 12L: 12D were the lowest, and the abundances of GSH-Px and CAT in the group of 8L: 16D were the highest. The abundance of SOD in the group of 8L: 16D was higher than that of the other two groups, and the abundance of SOD showed no significant difference between the other two groups (*p* > 0.05).

The abundances of antioxidant enzymes showed significant differences between the groups of 18 °C and 28 °C (*p* < 0.05). In the group of 18 °C, the abundance of MDA was higher and the abundances of SOD, CAT, and GSH-Px were lower. However, in the group of 28 °C, the abundance of MDA was lower and the abundances of SOD, CAT, and GSH-Px were higher.

### 3.2. Evaluation of Richness and Diversity of Bacterial Communities in Fish Guts

The results of the α-diversity analysis across different treatment groups are presented in Table 2. In terms of species richness, no significant differences were observed in the ACE and Chao1 indices among the three photoperiod groups (*p* > 0.05). The 18 °C group exhibited significantly higher ACE and Chao1 indices compared to the 28 °C group (*p* < 0.05). Regarding community diversity, there were still no significant differences among the light treatment groups, while in water temperature groups, the 18 °C group showed significantly higher Simpson and Shannon indices (*p* < 0.05). The Good’s Coverage exceeded 99% for all samples, indicating that the sequencing results reliably reflect the true microbial composition of the samples.

### 3.3. Composition, Relevance, and Species Abundance of Microbial Communities in Samples

Venn analysis of *S. scherzeri* in different photoperiod and water temperature groups were carried out (Figure 1). A total of 1958 and 1911 OTUs were identified from the samples of different photoperiod groups and water temperature groups, respectively. In the photoperiod groups, 1296 OTUs were shared across all the samples. The number of specific OTUs was the largest in the group of 12L: 12D (64) and the smallest in the group of 8L: 16D (50) (*p* < 0.05). With the increase in light duration, the number of specific OTUs firstly increased and then decreased. In the two water temperature groups, 1472 OTUs were shared in all samples. The number of unique OTUs in the group of 18 °C (253) was larger than that of the group of 28 °C (186).

The dominant microbial phyla in three photoperiod groups included Firmicutes, Proteobacteria, Acidobacteria, Bacteroidetes, Tenericutes, Actinobacteria, Chloroflexi, Gemmatimonadetes, Cyanobacteria, and Rokubacteria (Figure 2). Firmicutes was the most dominant phylum in the three photoperiod groups, and the abundances of Firmicutes in the three photoperiod groups were 32.1% (8L: 16D), 32.3% (12L: 12D), and 32.8% (16L: 8D). Proteobacteria was the second dominant microbial phylum, and the relative abundance of Proteobacteria in the group of 12L: 12D was significantly lower than that in other two groups (*p* < 0.05). Meanwhile, the relative abundance of Tenericutes also differed significantly, with the 12L: 12D group having a higher abundance than the other two groups. (*p* < 0.05).

The similarity analysis of microbial communities in the different photoperiod groups was also performed in the different water temperature groups. The dominant microbial phyla in these groups were *Firmicutes*, *Proteobacteria*, *Acidobacteria*, *Bacteroidetes*, *Actinobacteria*, *Chloroflexi*, *Gemmatimonadetes*, *Tenericutes*, *Rokubacteria*, and *Cyanobacteria* (Figure 3). *Firmicutes* and *Proteobacteria* were also the first and second dominant microbial phyla in the two groups. The relative abundance of *Tenericutes* showed the significant difference between the two temperature groups of 18 °C and 28 °C (*p* < 0.05).

Figure 4 shows the dominant microbial genera collected in three photoperiod groups. Lactobacillus was the highest abundant species in the groups of 8L: 16D (9.8%), 12L: 12D (10.6%) and 16L: 8D (10.3%). In addition, the dominant microbial genera in the group of 8L: 16D were *Ruminococcaceae_UCG-014* (6.5%), *Plesiomonas* (5.3%), *uncultured_bacterium_c_Subgroup_6* (3.6%), and *uncultured_bacterium_f_Muribaculaceae* (3.0%). In the samples in the group of 12L: 12D, dominant microbial genera were *Candidatus_Bacilloplasma* (10.2%), *Ruminococcaceae_UCG-014* (6.6%), *uncultured_bacterium_c_Subgroup_6* (4.6%), and *uncultured_bacterium_f_Muribaculaceae* (3.0%). In the samples in the group of 16L: 8D, dominant microbial genera were *Plesiomonas* (8.2%), *Ruminococcaceae_UCG-014* (6.5%), *Candidatus_Bacilloplasma* (4.1%), and *ncultured_bacterium_c_Subgroup_6* (3.1%).

Dominant microbial genera collected in different water temperature groups are shown in Figure 5. In the water temperature group of 18 °C, the dominant microbial genera were *Lactobacillus* (9.3%), *Ruminococcaceae_UCG-014* (6.4%), *uncultured_bacterium_c_Subgroup_6* (5.6%), *Candidatus_Bacilloplasma* (3.6%), and *uncultured_bacterium_f_Muribaculaceae* (2.7%). In the water temperature group of 28 °C, the dominant microbial genera were *Plesiomonas* (10.4%), *Lactobacillus* (9.5%), *Ruminococcaceae_UCG-014* (7.0%), *uncultured_bacterium_c_Subgroup_6* (3.6%), and *uncultured_bacterium_f_Muribaculaceae* (2.8%).

### 3.4. Similarity Analysis for Microbial Populations in Photoperiod Samples and Water Temperature Samples

Similarity analysis results of microbial genera were performed in the photoperiod (Figure 6) and water temperature samples (Figure 7). Samples in the groups of 8L: 16D and 16L: 8D were clustered into one category. Furthermore, this category was grouped with the 12L: 12D group into one sub-category (Figure 6). Some samples of the groups of 8L: 16D (G11) and 16L: 8D (G33) were clustered into one sub-category. The two water temperature groups were directly clustered into two sub-categories. The heatmap showed an opposite abundance in the microbial genera of the two groups (Figure 7).

## 4. Discussion

The determination of optimal fish farming conditions depends on assessing the environmental preferences of the fish, as indicated by variables that affect economic interests, such as health and growth [23]. The evaluation of antioxidant response is an effective way to assess the physiological state of fish and to reveal the physiological changes caused by intensive farming conditions [30,31]. An artificial environment can induce oxidative stress in the liver and affect the physiological state of fish [32,33]. The ability of antioxidant enzymes (SOD and CAT) and low molecular non-enzymatic antioxidants (GSH-Px), which remove free radicals to reduce oxidative damage to lipids, is involved in the antioxidant defense system [34,35,36]. Biological organisms are protected against molecular environmental stress through this ability to interact with antioxidants, which detoxify metabolites by converting them into water and oxygen [37,38,39,40]. MDA is a lipid peroxidation product that effectively represents the degree of oxidative damage in lipids [41,42]. High level of MDA contents is typically used as an indicator of antioxidant stress response. GSH-Px, as a crucial antioxidant in the body, is responsible for eliminating free radicals, purifying the internal environment, and regulating oxidative stress damage associated with disease development. The decrease in GSH-Px levels is a result of heightened demand and utilization of the tripeptide for lipid hydroperoxide metabolism [43]. The present study observed that both photoperiod and farmed water temperature affect the antioxidant defense system. For the photoperiod groups, MDA content was significantly higher in the group with a nearly natural photoperiod (12L: 12D) than in the groups with prolonged darkness (8L: 16D) and prolonged light duration (16L: 8D). This conclusion is corroborated by Malinovskyi et al., who proposed that a 12L: 12D photoperiod does not provide a dark phase duration sufficient for rest and recovery, nor does it offer a light phase long enough to enhance feed intake opportunities [11]. Therefore, this tendency led to increased per-oxidation and high MDA and low GSH-Px values under insufficient dark and light phases (12L: 12D). SOD activities in the present study were reduced in the liver of *S. scherzeri* in both the 12L: 12D and 16L: 8D photoperiod.

Elevated summer water temperatures have a detrimental effect on the welfare and production of fish [44]. Pereira et al. observed a significant interaction between photoperiod and temperature that caused oxidative stress in *Colossoma macropomum*, and oxidative stress levels increased markedly between 26 °C and 32 °C, regardless of the photoperiod [23]. Yang et al. reported that the cold-water fish *Schizothorax prenanti* acclimated at 11 °C revealed an optimal rearing temperature of 21 °C, as determined by SOD and CAT activity levels, as well as MDA contents [45]. In this sense, prolonged exposure appears to affect animals’ antioxidant responses in a specific manner. We observed an elevated MDA level associated with high water temperature (28 °C) in the present study. Liver GSH-Px levels were significantly higher in groups with higher water temperature (28 °C), indicating the key role of the liver in lipid metabolism. Prolonged high temperature exposure may adversely affect organism functions. In response, *S. scherzeri* may resist oxidative stress caused by high temperature stress by up-regulating the expression level of GSH-Px.

The SOD-CAT system serves as the primary defense mechanism against oxygen toxicity, with SOD facilitating the conversion of O_2_ and H_2_O_2_ and CAT contributing to transform H_2_O_2_ into water and oxygen [46,47]. Sun et al. found that SOD activity under a long light period was higher than that under a short light period in *Cynoglossus semilaevis* [48]. Prolonged light exposure caused the fish to accumulate a large amount of superpositive anions (O^2−^), which increase the concentration of O^2−^ in the blood. However, long-term accumulation of MDA in the liver inhibited SOD and CAT activities [49]. This mechanism leads to low SOD and CAT activities in the present study, with a negative trend in MDA contents in different photoperiod groups. This trend was in agreement with Malinovskyi et al. [11], who found that significantly higher SOD and CAT activities were found in liver of the 8L: 16D group compared to the 12L: 12D and 16L: 8D groups. Ma et al. found that the activity of GSH-Px in newly hatched *Takifugu rubripes* fish under 24L: 0D was significantly higher than in the other groups, suggesting that this increase may be a response to the impact of prolonged light exposure on larval cellular function [50]. Moreover, previous study has shown that high light intensity might cause oxidative stress in fish, representing a high activity of SOD with increased light intensity [51]. Thus, it should be noted that more research is needed to determine the antioxidant response in *S. scherzeri* to different light intensity stress.

During the 20 days of temperature treatments, SOD activities were upregulated under the heat stress in *S. scherzeri*. Some previous studies have reported the same conclusion as the present study [52]. However, CAT activity did not change during four weeks of hyperthermal stress, based on Chang’s report, which is compared with the current study [52]. CAT was upregulated as temperature increased (Table 1), suggesting that detoxification increased during this process, and the same results have been reported in previous studies [53,54]. In previous studies, low temperature has been shown to induce the upregulation of CAT activity in *Takifugu fasciatus* and *Takifugu fasciatus* [2,55]. These results might be caused by the different temperature tolerance of different species [52]. Compared to other species, larvae *S. scherzeri* require higher temperatures (25 °C to 28 °C) to maintain their physiology, resulting in high antioxidant enzymes activities as temperature was upregulated in the present study. In the process of farming, raising fish in continuous lighting and lower temperatures should be avoided.

Microbial communities in aquaculture water are crucial for the health and growth of fish, and fish also affect the composition of microbial communities in aquaculture water [56,57]. Different habitat environments have been shown to have different effects on the microbial composition of guts. In one study, as fish were acclimated to their surroundings, a community structure was developed according to their own physiological adaptations [58]. Larvae in their early development period have a significant response to light duration, resulting in variation in growth, development, reproduction, and metabolism. Puvanendran and Brown reported that a prolonged light exposure increased the growth rate of Atlantic cod [59]. The development of larvae, however, might be restrained by a prolonged photoperiod [60]. The composition of the fish gut microbiota is profoundly influenced by dietary patterns and food sources [61]. Under the optimal light duration, fish exhibited high activity levels, robust feeding behaviors, and rapid growth. Conversely, when the light intensity decreased below a critical threshold, it adversely affected the growth and development of fish, thus resulting in reduced activities, sluggish responses, and diminished feeding behavior [62]. Consequently, the impact of light on the fish gut microbiota might be indirectly mediated through its effects on diet. However, the abundance and diversity indices showed no significant differences among photoperiod groups in the present study. Walburn et al. observed that the most substantial shift in the microbiota of yellowtail kingfish (*Seriola lalandi*) larvae occurred during the dietary transition from live feed to formulated pellets, demonstrating that diet plays a pivotal role in microbial community development and assembly [63]. Thus, when a uniformly suitable temperature and identical food are maintained, variations in photoperiod leave the diversity of the gut microbiota in *S. scherzeri* unaffected.

Temperature directly affected the acquisition of fish foods, the activity of digestive enzymes, and the efficiency of food conversion [64]. Conversely, the effect of temperature on gut microbial abundance was significant, indicating a lower abundance corresponding to higher temperature in our study. The same conclusion is also supported by Zhou et al., who looked at rainbow trout under higher temperatures and found that the diversity of the intestinal microbiota decreased with an increase in temperature [65]. *Firmicutes* and *Proteobacteria* are the common predominant bacterial phyla associated with fish in both marine and freshwater environments [65,66,67,68,69]. The prevalence of *Proteobacteria* as a potential microbial marker of dysbiosis and disease in the gut microbiota might serve as an indicator of the health status of fish [70,71]. The diverse microorganisms encountered by cold-water fish experienced continual fluctuations due to the influences of various factors including seasonal changes, temperature variations, and environmental influences [72]. As a result, microbiota in guts showed a different relative abundance in photoperiod and farmed water temperature groups, especially *Proteobacteria* and *Tenericutes*. Other researchers indicated that dietary composition and host species played pivotal roles in shaping the diversity and structure of fish gut microbiota [73,74]. *Candidatus_Bacilloplasma* was considered as a symbiont (two organisms of different species with close and long-term biological interactions) and could be used as a potential taxonomic indicator to assess the health status of shrimps. The variations in photoperiod and water temperature had different influences on the growth, feeding behavior, and reproductive patterns of fish [75]. This result might be related to diet and host health impacting microbial composition, presenting variations observed at the genus level for *Candidatus_Bacilloplasma*. In the study, the relative abundance of *Plesiomonas* showed significant differences among different groups. *Plesiomonas shigelloides*, as an important pathogen in aquaculture, leads to a substantial mortality rate in diverse aquatic species [76]. The optimum growth temperature of *P. shigelloides* has been shown to be relatively high [77], thus resulting in the relatively high abundance in the group of 28 °C.

## 5. Conclusions

In the study, the antioxidant capacity and intestinal microbiota of *S. scherzeri* were explored under varying water temperatures and photoperiods. Photoperiod and temperature distinctly influenced the antioxidant capacity of mandarin fish. Both the 12L: 12D photoperiod at a constant temperature and the lower temperature of 18 °C induced antioxidant responses, creating suboptimal farming conditions. In contrast, while photoperiod had no impact on gut microbial diversity, it did significantly influence the relative abundance of microbial communities. An optimal high temperature could suppress oxidation in juvenile *S. scherzeri*. However, these conditions might influence the feeding behaviors of juvenile *S. scherzeri* and the composition of intestinal microbiota and promote the proliferation of opportunistic bacteria. In the process of farming, it should be avoided to raise fish in continuous lighting and lower temperatures.

## Figures and Tables

**Figure 1 biology-14-01400-f001:**
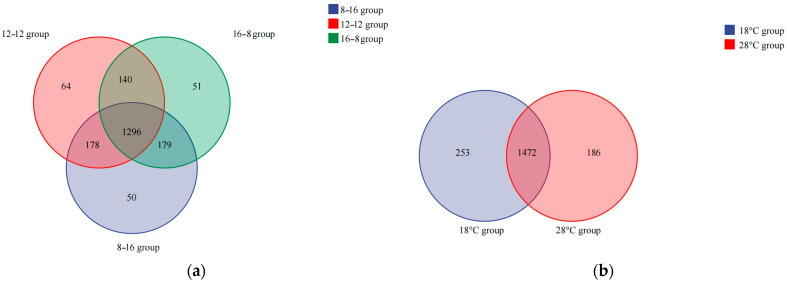
Venn diagram of shared and unique bacterial OTUs in three photoperiod groups (**a**) and two water temperature groups (**b**).

**Figure 2 biology-14-01400-f002:**
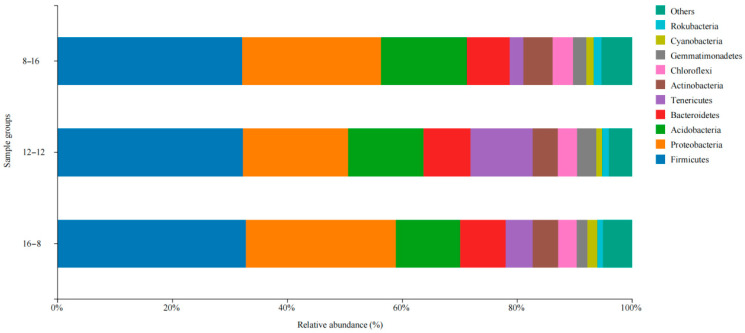
Microbial composition and abundance in the three photoperiod groups at the phylum.

**Figure 3 biology-14-01400-f003:**
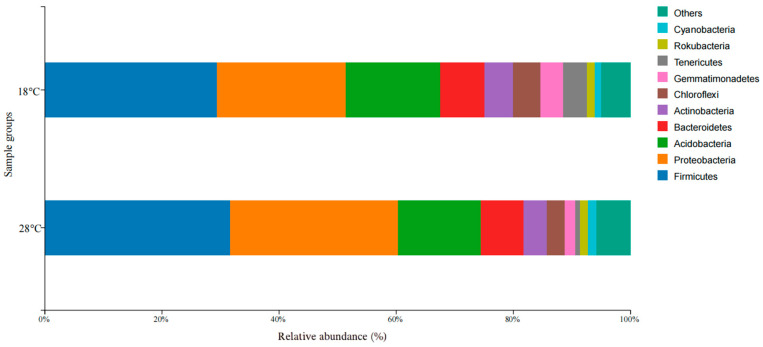
Microbial composition and abundance in *S. scherzeri* in two water temperature.

**Figure 4 biology-14-01400-f004:**
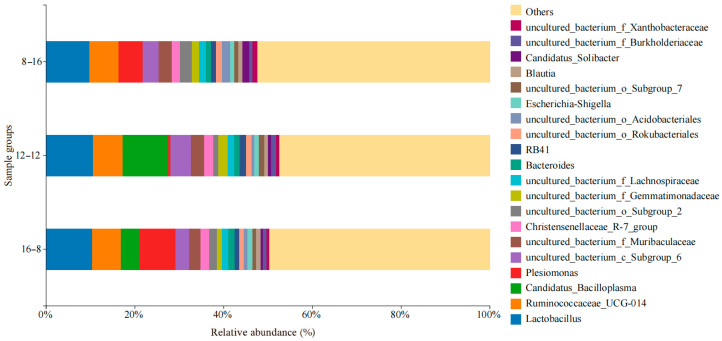
Microbial composition and abundance in *S. scherzeri* in three photoperiod groups at the genus level.

**Figure 5 biology-14-01400-f005:**
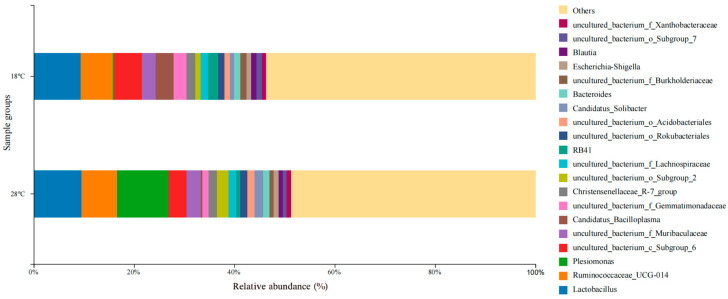
Microbial composition and abundance in *S. scherzeri* in two water temperature groups at the genus level.

**Figure 6 biology-14-01400-f006:**
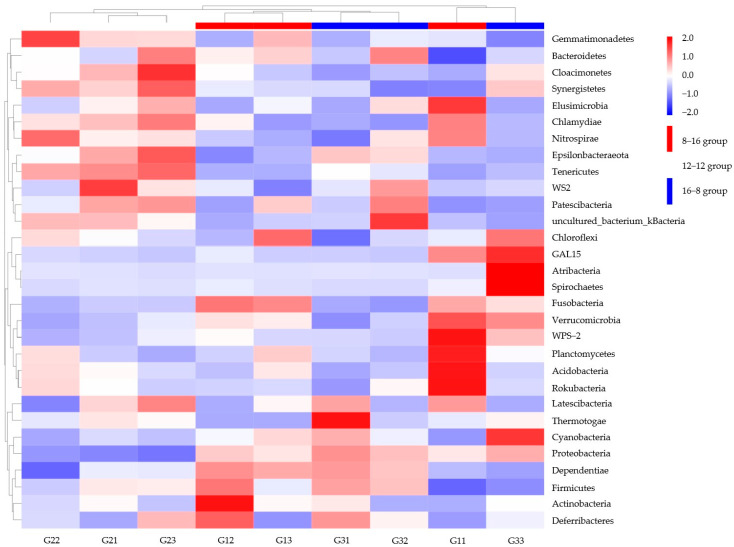
Heat map of *S. scherzeri* in different photoperiods at the phylum level. The color of the small rectangle indicates the abundance of bacteria in samples; red color indicates a high abundance, blue color indicates a low abundance. The group numbering is as follows: the letter “G” represents the photoperiod group. The first digit indicates the specific subgroup, while the second digit denotes the sample’s sequential number within that subgroup, e.g., G11 refers to the first sample in the first subgroup (8L: 16D) of the light group.

**Figure 7 biology-14-01400-f007:**
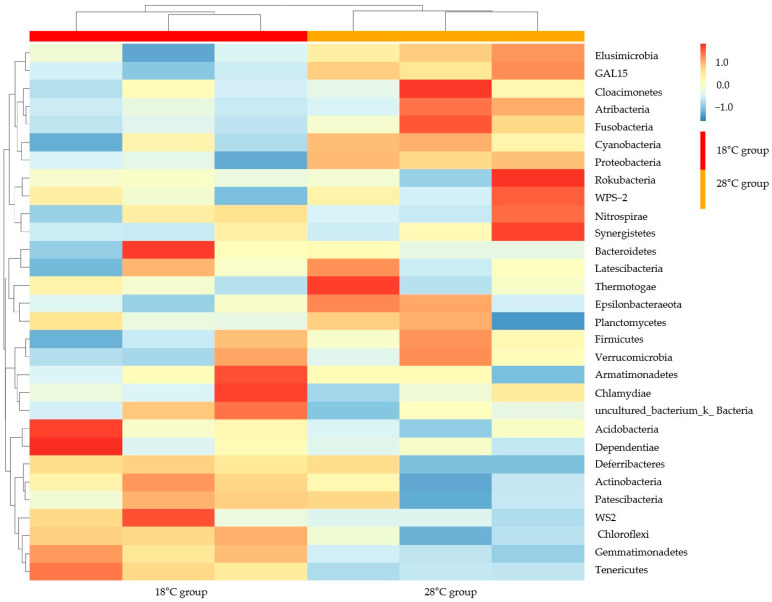
Heat map of *S. scherzeri* in different water temperatures at the phylum level. The color of the small rectangle indicates the abundance of bacteria in samples; red color indicates a high abundance, blue color indicates a low abundance.

**Table 1 biology-14-01400-t001:** Activities of antioxidant enzymes and MDA values of different groups.

Condition	Group	Antioxidant Enzymes			
		MDA (nmol/L)	SOD (U/mL)	GSH-Px (U/mL)	CAT (U/mL)
8L: 16D	G11	0.365	734.031	18.301	125.771
	G12	0.345	740.151	17.545	130.157
	G13	0.385	727.910	19.056	121.386
	average	0.365	734.031	18.301	125.771
	S.E.M	0.020	6.121	0.755	4.385
12L: 12D	G21	0.573	582.091	4.995	98.350
	G22	0.565	590.157	4.335	100.157
	G23	0.582	574.024	5.655	96.543
	average	0.573	582.091	4.995	98.350
	S.E.M	0.009	8.066	0.660	1.807
16L: 8D	G31	0.419	587.520	16.467	110.905
	G32	0.445	567.429	15.575	111.175
	G33	0.394	607.611	17.359	110.634
	average	0.419	587.520	16.467	110.905
	S.E.M	0.026	20.091	0.892	0.271
18 °C	W11	0.497	647.671	5.590	86.694
	W12	0.512	666.177	5.468	83.290
	W13	0.482	629.165	5.350	90.099
	average	0.497	647.671	5.468	86.694
	S.E.M	0.015	18.506	0.118	3.405
28 °C	W21	0.265	1029.465	11.476	146.258
	W22	0.276	1033.157	12.173	150.459
	W23	0.254	1025.773	10.780	142.058
	average	0.265	1029.465	11.476	146.258
	S.E.M	0.011	3.692	0.696	4.200

Notes: 8L: 16D represents 8 h of light and 16 h of darkness; 12L: 12D represents 12 h of light and 12 h of darkness; 16L: 8D represents 16 h of light and 8 h of darkness. The group numbering is as follows: the letter “G” represents the photoperiod group, and “W” represents the water temperature group. The first digit indicates the specific subgroup, while the second digit denotes the sample’s sequential number within that subgroup, e.g., G11 refers to the first sample in the first subgroup (8L: 16D) of the light group.

**Table 2 biology-14-01400-t002:** Abundance and diversity indices of species in different sampling groups at 97% similarity level.

Group	Observed OTUs	Predicted ACE	OTUs Chao1	Simpson	Shannon	Good’s Coverage
8L: 16D	1161.33 ± 106.82 ^a^	1210.84 ± 94.81 ^a^	1286.26 ± 86.91 ^a^	0.99 ± 0.00 ^a^	8.39 ± 0.27 ^a^	0.99 ± 0.00 ^a^
12L: 12D	1158.00 ± 19.67 ^a^	1240.04 ± 4.36 ^a^	1318.67 ± 5.91 ^a^	0.98 ± 0.00 ^a^	8.10 ± 0.11 ^a^	0.99 ± 0.00 ^a^
16L: 8D	1130.00 ± 82.82 ^a^	1195.09 ± 46.50 ^a^	1284.58 ± 50.49 ^a^	0.98 ± 0.01 ^a^	8.09 ± 0.35 ^a^	0.99 ± 0.00 ^a^
18 °C	1214.33 ± 77.59 ^a^	1272.36 ± 54.22 ^a^	1373.37 ± 32.52 ^a^	0.99 ± 0.00 ^a^	8.61 ± 0.09 ^a^	0.99 ± 0.00 ^a^
28 °C	1147.33 ± 33.56 ^a^	1193.52 ± 27.17 ^b^	1268.18 ± 62.31 ^b^	0.98 ± 0.00 ^a^	8.23 ± 0.15 ^b^	0.99 ± 0.00 ^a^

Notes: 8L: 16D represents 8 h of light and 16 h of darkness; 12L: 12D represents 12 h of light and 12 h of darkness; 16L: 8D represents 16 h of light and 8 h of darkness. Data are presented as mean ± S.E.M. The column exhibiting disparate characters denotes statistically significant variations among the groups, as determined by a one-way analysis of variance (ANOVA) coupled with a subsequent post hoc test (*p* < 0.05).

## Data Availability

The data that support the findings of this study are available from the corresponding author upon reasonable request.

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
