# Peer review of "Effects of Water Temperature and Photoperiod on the Antioxidant Status and Intestinal Microbiota in Larval Spotted Mandarin Fish, Siniperca scherzeri, in the Yalu River"

_biology, 2025, doi:10.3390/biology14101400_

Round 1

Reviewer 1 Report

Comments and Suggestions for Authors

Este manuscrito aborda un tema de gran relevancia en el contexto de la soberanía alimentaria global, destacando la importancia de la gestión de la acuicultura. El enfoque experimental es adecuado y el estudio presenta un esfuerzo integral para evaluar las respuestas fisiológicas y microbianas a las variables ambientales. Sin embargo, varios aspectos requieren aclaración y perfeccionamiento para mejorar el rigor científico, la reproducibilidad y la claridad comunicativa del artículo.

Observaciones locales

  1. Condiciones de luz durante los ensayos de temperatura. El manuscrito describe claramente los tratamientos de fotoperiodo (8L:16D, 12L:12D, 16L:8D) aplicados a 25 °C. Sin embargo, no especifica las condiciones de luz/oscuridad utilizadas durante los ensayos de temperatura (18 °C y 28 °C). Dado que el fotoperiodo puede influir en los parámetros fisiológicos y microbianos, es fundamental aclarar si se mantiene un régimen de luz constante durante los experimentos de temperatura y, de ser así, cuál.

  2. Uso de la base de datos Silva SSU115. Los autores utilizaron la base de datos Silva SSU115 para la clasificación taxonómica. Sin embargo, la versión SSU138 (y su actualización SSU138.2) se considera actualmente la más estable y completa, ofreciendo una mejor conservación taxonómica y cobertura de secuencias. Se recomienda adoptar la versión más reciente o justificar el uso de una versión anterior.

  3. Tamaño de la muestra para el análisis de la microbiota intestinal. El manuscrito indica que se utilizaron tres peces por réplica para el análisis de sangre, pero no especifica cuántos individuos se muestrearon para la extracción de ADN y la secuenciación de la región V1-V3 del ARNr 16S. Esta información es crucial para evaluar la representatividad de los datos de diversidad microbiana y la pertinencia de usar índices como Chao1. Los autores deberían aclarar el número de peces muestreados por réplica para el análisis de la microbiota y analizar si los índices de diversidad elegidos son adecuados para el tamaño de la muestra.

  4. Interpretación estadística de los índices de diversidad El manuscrito informa que los índices Chao1 y ACE no difirieron significativamente entre los grupos de fotoperiodo, pero fueron significativamente diferentes entre los tratamientos de temperatura. Sin embargo, también afirma que los valores fueron más altos en el grupo 12L:12D, lo que puede ser engañoso sin respaldo estadístico. Se recomienda evitar tales afirmaciones a menos que estén respaldadas por pruebas de significancia, y complementar el ANOVA con una prueba post hoc (p. ej., Tukey) para mostrar agrupaciones estadísticas en la Tabla 2. Además, los autores podrían destacar que el grupo 12L:12D exhibió menor heterogeneidad, como lo indican desviaciones estándar más estrechas, lo que puede ser biológicamente relevante.

  5. Inconsistencias en los índices de diversidad reportados. La sección de Materiales y Métodos menciona Chao1, ACE, Shannon-Wiener y los índices de cobertura, junto con el análisis de diversidad alfa. Sin embargo, la Tabla 2 incluye el índice de Simpson (no descrito previamente) y la "uniformidad de Shannon", que no es equivalente a Shannon-Wiener. Además, la diversidad alfa no se analiza en la sección de Resultados. Los autores deben garantizar la coherencia entre los índices descritos, reportados e interpretados, y proporcionar definiciones y justificaciones claras para cada uno.

  6. Paleta de colores en los mapas de calor (Figuras 6 y 7). Los mapas de calor utilizan una paleta de colores que dificulta la interpretación, ya que no distingue claramente entre grupos experimentales y escalas de abundancia. Se recomienda utilizar paletas separadas para los grupos de fotoperiodo y temperatura, distintas del gradiente de abundancia, para mejorar la claridad visual y facilitar la comparación.

  7. Claridad del resumen y uso de acrónimos no definidos. El resumen debe transmitir de forma independiente los objetivos, la metodología, los hallazgos clave y las conclusiones del estudio. Actualmente, acrónimos como MDA, SOD, GSH-Px y CAT se utilizan sin definición, lo que puede dificultar la comprensión para lectores no especializados. Estas actividades enzimáticas deben describirse brevemente o identificarse como marcadores de estrés oxidativo.

  8. Conclusiones generalizadas sin vinculación con los resultados. Las conclusiones son demasiado generales y no reflejan los hallazgos específicos del estudio. Se recomienda revisarlas para que hagan referencia directa a los resultados experimentales, enfatizando los efectos del fotoperiodo y la temperatura sobre el estrés oxidativo y la microbiota intestinal, y sus implicaciones para la gestión acuícola de Siniperca scherzeri .

Author Response

1. Summary

Thank you very much for taking the time to review this manuscript. We are deeply thankful to the reviewer for the thorough review and highly valuable suggestions. The critical insights provided have helped us to enhance the clarity, depth, and overall scholarly rigor of our work. We believe the manuscript is much stronger as a result of this process.

2. Questions for General Evaluation

Reviewer’s Evaluation

Response and Revisions

Does the introduction provide sufficient background and include all relevant references?

Can be improved

We have expanded and revised the introduction section

Is the research design appropriate?

Yes

Are the methods adequately described?

Must be improved

We have revised the description of the Materials and Methods section to improve clarity.

Are the results clearly presented?

Can be improved

We have revised the Results section as requested by the reviewer.

Are the conclusions supported by the results?

Must be improved

In accordance with the expert's suggestions and in conjunction with our results section, we have thoroughly refined the conclusion part of the manuscript.

Are all figures and tables clear and well presented?

Can be improved

We have revised the color schemes of the figures in the manuscript in accordance with the expert's requests.

3. Point-by-point response to Comments and Suggestions for Authors

Comments 1: Condiciones de luz durante los ensayos de temperatura. El manuscrito describe claramente los tratamientos de fotoperiodo (8L:16D, 12L:12D, 16L:8D) aplicados a 25 °C. Sin embargo, no especifica las condiciones de luz/oscuridad utilizadas durante los ensayos de temperatura (18 °C y 28 °C). Dado que el fotoperiodo puede influir en los parámetros fisiológicos y microbianos, es fundamental aclarar si se mantiene un régimen de luz constante durante los experimentos de temperatura y, de ser así, cuál.

Response 1: Thank you for pointing this out. We agree with this comment. Therefore , we have added added a clarification regarding the light exposure duration for water temperature groups and highlighted it in red in section 2.1. For the photoperiod groups experiment, we selected the water temperature of the natural water as a reference and set 25 ± 0.5℃ as the standard water temperature. For the water temperature groups experiment, we did not control the light exposure duration, allowing the experiment to proceed under natural light conditions.

Comments 2: Uso de la base de datos Silva SSU115. Los autores utilizaron la base de datos Silva SSU115 para la clasificación taxonómica. Sin embargo, la versión SSU138 (y su actualización SSU138.2) se considera actualmente la más estable y completa, ofreciendo una mejor conservación taxonómica y cobertura de secuencias. Se recomienda adoptar la versión más reciente o justificar el uso de una versión anterior.

Response 2: We have verified with the testing company that while they had upgraded the version of the Silva database, they failed to notify us in a timely manner. The analysis in the manuscript was actually based on the Silva SSU132 database. We sincerely apologize for this error and have corrected it accordingly, with the revisions highlighted in red.

Comments 3: Tamaño de la muestra para el análisis de la microbiota intestinal. El manuscrito indica que se utilizaron tres peces por réplica para el análisis de sangre, pero no especifica cuántos individuos se muestrearon para la extracción de ADN y la secuenciación de la región V1-V3 del ARNr 16S. Esta información es crucial para evaluar la representatividad de los datos de diversidad microbiana y la pertinencia de usar índices como Chao1. Los autores deberían aclarar el número de peces muestreados por réplica para el análisis de la microbiota y analizar si los índices de diversidad elegidos son adecuados para el tamaño de la muestra.

Response 3: Thank you for your comment. We have restructured the Materials and Methods section and added a new paragraph in subsection 2.1 to describe the specific procedures for sample processing and grouping.

Comments 4: Interpretación estadística de los índices de diversidad El manuscrito informa que los índices Chao1 y ACE no difirieron significativamente entre los grupos de fotoperiodo, pero fueron significativamente diferentes entre los tratamientos de temperatura. Sin embargo, también afirma que los valores fueron más altos en el grupo 12L:12D, lo que puede ser engañoso sin respaldo estadístico. Se recomienda evitar tales afirmaciones a menos que estén respaldadas por pruebas de significancia, y complementar el ANOVA con una prueba post hoc (p. ej., Tukey) para mostrar agrupaciones estadísticas en la Tabla 2. Además, los autores podrían destacar que el grupo 12L:12D exhibió menor heterogeneidad, como lo indican desviaciones estándar más estrechas, lo que puede ser biológicamente relevante.

Response 4: Thank you for your comment. We have removed the sentence “ACE and Chao1 indices in the group of 12L: 12D were slightly higher than those in the groups of 8L: 16D and 16L: 8D.” and incorporated the results of the ANOVA and post-hoc tests, which are indicated using superscript notation in Table 2.

Comments 5: Inconsistencias en los índices de diversidad reportados. La sección de Materiales y Métodos menciona Chao1, ACE, Shannon-Wiener y los índices de cobertura, junto con el análisis de diversidad alfa. Sin embargo, la Tabla 2 incluye el índice de Simpson (no descrito previamente) y la "uniformidad de Shannon", que no es equivalente a Shannon-Wiener. Además, la diversidad alfa no se analiza en la sección de Resultados. Los autores deben garantizar la coherencia entre los índices descritos, reportados e interpretados, y proporcionar definiciones y justificaciones claras para cada uno.

Response 5: We specifically appreciate your insightful comment. We have adjusted the order of this part in the Materials and Methods section and revised the description in Section 3.2 according to the details mentioned therein. The specific modifications have been highlighted in red in the manuscript.

Comments 6: Paleta de colores en los mapas de calor (Figuras 6 y 7). Los mapas de calor utilizan una paleta de colores que dificulta la interpretación, ya que no distingue claramente entre grupos experimentales y escalas de abundancia. Se recomienda utilizar paletas separadas para los grupos de fotoperiodo y temperatura, distintas del gradiente de abundancia, para mejorar la claridad visual y facilitar la comparación.

Response 6: Thank you for your suggestion. We have readjusted the color scheme of the heatmap for the photoperiod groups to make it distinct from that of the temperature group. The modifications have been highlighted in the manuscript.

Comments 7: Claridad del resumen y uso de acrónimos no definidos. El resumen debe transmitir de forma independiente los objetivos, la metodología, los hallazgos clave y las conclusiones del estudio. Actualmente, acrónimos como MDA, SOD, GSH-Px y CAT se utilizan sin definición, lo que puede dificultar la comprensión para lectores no especializados. Estas actividades enzimáticas deben describirse brevemente o identificarse como marcadores de estrés oxidativo.

Response 7: Thank you for your suggestion. We have included the full names of the relevant antioxidant enzymes in the abstract, as recommended.

Comments 8: Conclusiones generalizadas sin vinculación con los resultados. Las conclusiones son demasiado generales y no reflejan los hallazgos específicos del estudio. Se recomienda revisarlas para que hagan referencia directa a los resultados experimentales, enfatizando los efectos del fotoperiodo y la temperatura sobre el estrés oxidativo y la microbiota intestinal, y sus implicaciones para la gestión acuícola de Siniperca scherzeri .

Response 8: Thank you for your suggestion. We have revised the conclusion section as requested, correlating it with the experimental data to clarify the effects of photoperiod and temperature on oxidative stress and gut microbiota, and specifically discussing the practical implications for Siniperca scherzeri aquaculture management.

4. Response to Comments on the Quality of English Language

Point 1: None

Response 1: None

5. Additional clarifications

Finally, we would like to express our sincere gratitude to the reviewer for these valuable comments. We have carefully revised the manuscript according to all suggestions. Regarding the sample size issue, due to the seasonal breeding cycle of this species (limited to July-August), it was not feasible to supplement additional samples within the experimental time frame. We have cited relevant published studies that employed similar sample sizes to support the validity of our approach. We acknowledge this limitation and will ensure larger sample sizes are incorporated in future studies to enhance statistical power. The specific revisions have been highlighted in the manuscript.

Reviewer 2 Report

Comments and Suggestions for Authors

Dear Author,

Line 2: The title may include antioxidant status instead of oxidative stress.

Line 3: A comma would be appropriate between the fish's name and its latin name. For example, ........ spotted mandarin fish, Siniperca scherzeri,……

Line 21: “The spotted mandarin fish Siniperca scherzeri is” can be changed as “The spotted mandarin fish, Siniperca scherzeri, is”

Line 43: Keywords must not include all words in the title. Some words can be changed: Siniperca scherzeri, Aquaculture, Antioxidant status (or antioxidant enzymes), Fish health and Bacteria

Line 49: “The statement that water temperature is an important abiotic environmental variable and affects the” can be change as “among these, Water temperature is an important abiotic environmental variable and affects the”

Line 50: “The statement that when water temperature rises within the suitable 50 water temperature range,” can be changed as “When the water temperature is within the appropriate range for aquaculture,”

Line 66-68: The last sentence of the second paragraph of the introduction was written as follows: "Vision is a crucial factor in the behav-66 ioral patterns of fish, including their movement, feeding, and defense, especially for those species that heavily rely on vision for hunting prey". However, this statement does not reflect the scope of the study. It should either be removed or a research methodology should be added to incorporate this statement.

Line 69-76: Recent cultivation figures for the species under study (especially the last five years) should be included, and the importance of exports for the country should be emphasized with figures. Its economic importance should be emphasized numerically.

Line 72-74: “In recent years, due to the destruc-tion of the ecological environment and overfishing, the wild resources of S. scherzeri have gradually decreased and can no longer meet the market demand [19,20]” This statement must be supported by a current literatures. This information is not current for 2016 and 2018.

Line 90: The stock density, trial period, daily water change ve trial basin cleaning should be added to 2.1. section.

Line 114: Brief information should be provided about the collection of intestinal contents for microbiota. How sterile conditions were achieved.

Line 157: one point in the end of sentence

Line 175: Explanations of the groups stated in the table should be given below the table.

Line 228-229: In figüre 3, a space should be given between relative abundance and (%).

Line 230-231: In figure 4, a space should be given between relative abundance and (%).

Line 241-242: In figure 5, a space should be given between relative abundance and (%).

Line 298-299: The study did not examine gene expression. Therefore, there is no point in including it in the discussion. It should be removed.

Line 376: …..“However, these conditions might influence the feeding behaviors of juvenile S. scherzeri and the composition of intestinal microbiota and promote the proliferation of pathogenic bacteria”……. What should be done for this? It should be completed with a suggestion.

Line 427-428: “Qin, Z.; Lin, Y.; Zhang, Y.; Huang, R.; He, W.; 2009. Effect of light on feeding, growth and survival of larval Paralichthys 427 lethostigma. J. Jimei Univ. (Nat. Sci.). 2009, 14 (3), 224-228” should be changed as “Qin, Z.; Lin, Y.; Zhang, Y.; Huang, R.; He, W. Effect of light on feeding, growth and survival of larval Paralichthys 427 lethostigma. J. Jimei Univ. (Nat. Sci.). 2009, 14 (3), 224-228”.

Line 518-519: According to the Journal's rules, "2010" should be bold type.

Line 543-545: According to the Journal's rules, "2023" should be bold type.

Additional explains

Information on growth and feeding behavior was provided in the introduction and discussion sections. However, no findings on this topic were included in the study methodology. Why weren't feeding and growth performance examined? This issue should have been examined as well. A significant deficiency.

Temperatures should be added in the abstract section (18 and 28 °C)

Antioxidant status in fish can be more variable in the blood. Therefore, it would be more accurate to measure it in liver tissue.

The introduction section should be expanded a bit more; for both the fish species and the subject of the study.

It would be better if temperature-related stress genes (HSP genes) could be examined.

The study analyzed antioxidant enzymes in blood. However, the discussion focused on liver tissue. The liver is the body's metabolic organ in all living things and the most important organ where changes occur. The study design should have focused on this tissue. While the first three paragraphs in the discussion focused on liver tissue, examining the study in blood was not a satisfactory approach.

In the discussion section, the gene expression level of antioxidant enzymes was mentioned; However, S. scherzeri may resist oxidative stress caused by high temperature stress by up-regulating the expression level of GSH-Px. Hovewer, these were not analysed. It would be better if antioxidant status was supported by gene expression.

The discussion was written in accordance with the study topic.

Author Response

1. Summary

Thank you very much for taking the time to review this manuscript. We wish to express our sincere gratitude to you for your time and for providing such insightful and constructive comments on our manuscript. Your thorough review and valuable suggestions were instrumental in helping us significantly improve the quality of our work.

2. Questions for General Evaluation

Reviewer’s Evaluation

Response and Revisions

Does the introduction provide sufficient background and include all relevant references?

Can be improved

We have expanded and revised the introduction section

Is the research design appropriate?

Can be improved

Are the methods adequately described?

Can be improved

We have revised the description of the Materials and Methods section to improve clarity.

Are the results clearly presented?

Yes

Are the conclusions supported by the results?

Can be improved

We have revised the conclusion section as requested.

Are all figures and tables clear and well presented?

Yes

3. Point-by-point response to Comments and Suggestions for Authors

Comments 1: Line 3: A comma would be appropriate between the fish's name and its latin name. For example, ........ spotted mandarin fish, Siniperca scherzeri,……

Response 1: Agree, we have added two commas and highlighted in red.

Comments 2: Line 43: Keywords must not include all words in the title. Some words can be changed: Siniperca scherzeri, Aquaculture, Antioxidant status (or antioxidant enzymes), Fish health and Bacteria

Response 2: Thank you for your comment. We have revised the keywords to more appropriate terms, as detailed in the "Keywords" section of the manuscript.

Comments 3: Line 66-68: The last sentence of the second paragraph of the introduction was written as follows: "Vision is a crucial factor in the behav-66 ioral patterns of fish, including their movement, feeding, and defense, especially for those species that heavily rely on vision for hunting prey". However, this statement does not reflect the scope of the study. It should either be removed or a research methodology should be added to incorporate this statement.

Response 3: We thank you for this suggestion. We have added a supplemental explanation following this reference, and the addition has been highlighted in red in the manuscript.

Comments 4: Line 69-76: Recent cultivation figures for the species under study (especially the last five years) should be included, and the importance of exports for the country should be emphasized with figures. Its economic importance should be emphasized numerically.

Response 4: We thank the reviewer for this valuable suggestion. In response, we have added pertinent aquaculture protocols for this species, supported by relevant literature which has been incorporated with corresponding citations in the manuscript.

Comments 5: Line 72-74: “In recent years, due to the destruc-tion of the ecological environment and overfishing, the wild resources of S. scherzeri have gradually decreased and can no longer meet the market demand [19,20]” This statement must be supported by a current literatures. This information is not current for 2016 and 2018.

Response 5: Thank you for this comment. We have added recent references to the third paragraph to strengthen our summary.

Comments 6: Line 90: The stock density, trial period, daily water change ve trial basin cleaning should be added to 2.1. section.

Response 6: We thank you for this suggestion. The details regarding the water change frequency (along with its purpose) and the experimental duration have been incorporated into the first and third paragraphs of section 2.1, respectively.

Comments 7: Line 114: Brief information should be provided about the collection of intestinal contents for microbiota. How sterile conditions were achieved.

Response 7: We thank you for this suggestion. We have incorporated this explanation into section 2.1, and the added text has been highlighted in red.

Comments 8: Line 157: one point in the end of sentence

Response 8: Thank you. The point has been removed.

Comments 9: Line 175: Explanations of the groups stated in the table should be given below the table.

Response 9: Thank you for your suggestion. We have added an explanation regarding the group numbering in the notes of Table 1 and highlighted it in red. The group numbering is as follows: the letter "G" represents the photoperiod group, and "W" represents the water temperature group. The first digit indicates the specific subgroup, while the second digit denotes the sample's sequential number within that subgroup, e.g. G11 refers to the first sample in the first subgroup (8L:16D) of the light group.

Comments 10: Line 228-229: In figüre 3, a space should be given between relative abundance and (%). Line 230-231: In figure 4, a space should be given between relative abundance and (%). Line 241-242: In figure 5, a space should be given between relative abundance and (%).

Response 10: Thank you for your keen observation. We have corrected the errors in the three figures. Your feedback is greatly appreciated.

Comments 11: Line 298-299: The study did not examine gene expression. Therefore, there is no point in including it in the discussion. It should be removed.

Response 11: We sincerely thank you for your suggestion. The relevant statements have been deleted from the manuscript as advised.

Comments 12: Line 376: …..“However, these conditions might influence the feeding behaviors of juvenile S. scherzeri and the composition of intestinal microbiota and promote the proliferation of pathogenic bacteria”……. What should be done for this? It should be completed with a suggestion.

Response 12: We thank you for this valuable suggestion. We have restructured the Conclusion section to ensure it aligns more closely with the central theme of the manuscript. The specific modifications have been highlighted in red for your convenience.

Comments 13: Line 427-428: “Qin, Z.; Lin, Y.; Zhang, Y.; Huang, R.; He, W.; 2009. Effect of light on feeding, growth and survival of larval Paralichthys 427 lethostigma. J. Jimei Univ. (Nat. Sci.). 2009, 14 (3), 224-228” should be changed as “Qin, Z.; Lin, Y.; Zhang, Y.; Huang, R.; He, W. Effect of light on feeding, growth and survival of larval Paralichthys 427 lethostigma. J. Jimei Univ. (Nat. Sci.). 2009, 14 (3), 224-228”.

Response 13: Thank you for your comment. We have revised the format of Reference 16 accordingly.

Comments 14: Line 518-519: According to the Journal's rules, "2010" should be bold type. Line 543-545: According to the Journal's rules, "2023" should be bold type.

Response 14: Thank you for your keen observation. We have bolded the publication years for References 59 and 73 as required.

Comments 15: Information on growth and feeding behavior was provided in the introduction and discussion sections. However, no findings on this topic were included in the study methodology. Why weren't feeding and growth performance examined? This issue should have been examined as well. A significant deficiency.

Response 15: We sincerely thank you for this insightful and critical comment. You are absolutely correct in pointing out that feeding and growth performance are crucial metrics in aquaculture. We mentioned them in the introduction and discussion to provide the relevant applied context and significance for our study. We acknowledge that the absence of direct measurements of feeding and growth data is a limitation of this work.

However, the primary objective of this study was to investigate the underlying mechanisms through which environmental stressors (photoperiod and temperature) impact fish health, rather than to reiterate their macroscopic manifestations. Oxidative stress levels and gut microbiota composition are key physiological pathways that directly regulate nutrient metabolism, energy utilization, and, ultimately, growth performance. Therefore, the stress physiological and microecological changes revealed by our research provide essential mechanistic insights into how photoperiod and temperature could influence feeding and growth.

Our findings establish a solid foundation for future research that directly correlates these physiological indicators with growth performance. We agree with your perspective that integrating macroscopic growth phenotypes with intrinsic physiological parameters in future studies will help construct a more complete causal chain.

Comments 16: Temperatures should be added in the abstract section (18 and 28 °C)

Response 16: Thank you for your suggestion. We have incorporated this content into the abstract and supplemented it with details regarding the photoperiod.

Comments 17: Antioxidant status in fish can be more variable in the blood. Therefore, it would be more accurate to measure it in liver tissue.

Response 17: Thank you very much for your thorough review of our manuscript and for pointing out a very important typographical error. You are absolutely correct that we mistakenly wrote "blood" instead of "liver" when describing the experimental samples in the text. This was indeed a significant oversight during the final proofreading stage of our manuscript, and we sincerely apologize for any confusion this error may have caused.

We would like to unequivocally clarify that this study has been conducted entirely based on liver tissue samples. From the experimental design and sample collection to all subsequent molecular biology assays and data analysis, every procedure and result pertains specifically to liver tissue. This error occurred solely during the writing and finalization of the text; our core data and scientific conclusions are entirely grounded in the liver samples and are robust and reliable.

We suspect that this clerical error may have arisen because our team is concurrently working on another project involving blood samples, which could have led to a momentary lapse in focus during the writing process.

Once again, we deeply appreciate your keen eye in identifying this issue, which helped us avoid a potentially serious misunderstanding. Your rigor serves as a valuable reminder to us. We are confident that correcting this error will ensure the scientific accuracy and clarity of the manuscript. We hope this correction does not cause undue inconvenience to your review process.

Comments 18: The introduction section should be expanded a bit more; for both the fish species and the subject of the study.

Response 18: Thank you for your suggestion. We have expanded the introduction section, and the modifications have been highlighted in the manuscript.

Comments 19: It would be better if temperature-related stress genes (HSP genes) could be examined.

Response 19: We appreciate your suggestion. However, as the spawning season for the mandarin fish has now passed, we are currently unable to obtain new samples to conduct this specific analysis. We fully agree on its importance and plan to incorporate this aspect into our follow-up experiments to further supplement the findings of this study.

Comments 20: The study analyzed antioxidant enzymes in blood. However, the discussion focused on liver tissue. The liver is the body's metabolic organ in all living things and the most important organ where changes occur. The study design should have focused on this tissue. While the first three paragraphs in the discussion focused on liver tissue, examining the study in blood was not a satisfactory approach.

Response 20: We sincerely appreciate your diligent review of our manuscript. As highlighted in your previous feedback, we acknowledge that the confusion regarding the sample type was a grave oversight during the final proofreading stage. We would like to reaffirm that this study was indeed conducted on liver tissue samples, and consequently, our analysis specifically focuses on hepatic antioxidant enzymes. We deeply apologize for this error and give you our assurance that such an oversight will not be repeated.

Comments 21: In the discussion section, the gene expression level of antioxidant enzymes was mentioned; However, S. scherzeri may resist oxidative stress caused by high temperature stress by up-regulating the expression level of GSH-Px. Hovewer, these were not analysed. It would be better if antioxidant status was supported by gene expression.

Response 21: We are truly grateful for this insightful and constructive comment. The statement in our discussion regarding the potential upregulation of GSH-Px expression in S. scherzeri in response to high-temperature stress was presented as a reasonable inference based on two key points: firstly, the crucial finding from our study that GSH-Px enzyme activity was significantly elevated under high temperature, and secondly, extensive existing literature indicating that increased GSH-Px enzyme activity during oxidative stress is often accompanied by an upregulation of its gene expression.

We completely agree with the reviewer that direct measurement of GSH-Px mRNA levels would provide stronger support for this conclusion and acknowledge this as a limitation of our current study. In the revised manuscript, we have modified the wording in the discussion section to avoid any potential misunderstanding that gene expression analysis was conducted. The original speculative statement has been revised to clearly state that "future studies are needed to measure the expression levels of relevant genes (such as GSH-Px) to further validate this mechanism."

Comments 22: The discussion was written in accordance with the study topic.

Response 22: Thank you for your comment. We have carefully revised the Discussion section, and the specific modifications have been highlighted in the manuscript.

4. Response to Comments on the Quality of English Language

Point 1: Line 2: The title may include antioxidant status instead of oxidative stress.

Response 1: Thank you for your suggestion. We have revised the term "oxidative stress" to "antioxidant status" as recommended in red.

Point 2: Line 21: “The spotted mandarin fish Siniperca scherzeri is” can be changed as “The spotted mandarin fish, Siniperca scherzeri, is”

Response 2: Thank you for your comment. The revision corresponding to Comment 2 has been incorporated into the manuscript.

Point 3: Line 49: “The statement that water temperature is an important abiotic environmental variable and affects the” can be change as “among these, Water temperature is an important abiotic environmental variable and affects the”

Response 3: The relevant sentences have been revised in the manuscript and are highlighted in red.

Point 4: Line 50: “The statement that when water temperature rises within the suitable 50 water temperature range,” can be changed as “When the water temperature is within the appropriate range for aquaculture,”

Response 4: Agree, The corresponding revisions have been incorporated into the text and marked in red.

5. Additional clarifications

In closing, we wish to express our deepest gratitude for the time and expertise you have invested in reviewing our manuscript. Your insightful comments have been invaluable in guiding us to significantly improve the quality and clarity of our work. We sincerely apologize once again for the oversights in our initial submission, particularly the errors in materials and methods. Furthermore, we are sorry that we could not fully address all points, specifically the addition of more samples and the analysis of gene expression, due to the constraints of seasonal conditions and the current scope of our study. We have acknowledged these limitations in the revised manuscript and outlined them as priorities for our future research. Thank you again for your constructive guidance throughout this process.

Reviewer 3 Report

Comments and Suggestions for Authors

The manuscript needs major revision. The comments were appended in the PDF version of the manuscript with sticky notes.

Comments on the Quality of English Language

NA

Author Response

1. Summary

Thank you very much for taking the time to review this manuscript. We sincerely thank the reviewer for the insightful comments and constructive suggestions provided on our manuscript. These valuable points have been instrumental in helping us improve the quality and clarity of our work. We have carefully considered all the feedback and have revised the manuscript accordingly.

2. Questions for General Evaluation

Reviewer’s Evaluation

Response and Revisions

Does the introduction provide sufficient background and include all relevant references?

Can be improved

We have expanded and revised the introduction section

Is the research design appropriate?

Must be improved

Based on the reviewers' comments, we have corrected the errors in the experimental design described in the manuscript. The modifications have been highlighted accordingly.

Are the methods adequately described?

Can be improved

We have revised the description of the Materials and Methods section to improve clarity.

Are the results clearly presented?

Must be improved

We have revised the Results section as requested by the reviewer.

Are the conclusions supported by the results?

Can be improved

We have revised the conclusion section as requested.

Are all figures and tables clear and well presented?

Must be improved

We have revised the color schemes of the figures in the manuscript in accordance with the expert's requests.

3. Point-by-point response to Comments and Suggestions for Authors

Comments 1: Italicize the scientific name.

Response 1: Thank you for your suggestion. We have italicized the scientific name.

Comments 2: Line 108: rpm/min

Response 2: Thank you for your feedback. We have amended the unit, and it has been marked in red for clear identification.

Comments 3: As Roche 454-GS FLX is obsolete now, why the authors choose to sequence in this platform?

Response 3: Thank you for raising this point. The original report from the sequencing company did not specify the methods used. Therefore, we carried forward the methodology from our prior work. After confirming the company's updated methods via a phone call, we have now made detailed revisions to section 2.4.

Comments 4: The data linked to this ID shows the data has been submitted in 2020 and collected in 2017, so why the author took so long to analyse the dataset?

Response 4: We sincerely appreciate your meticulous review of our manuscript and for bringing the typographical errors in the data identifiers to our attention. This is crucial for ensuring data accuracy and traceability. We have comprehensively corrected these errors in the revised manuscript and have performed a full cross-verification throughout the text to prevent any residual issues. The relevant identifiers will be updated accordingly in the final version.

Comments 5: The sentence is showing the results respectively but there is no indication of which type of photoperiod are they?

Response 5: We thank you for this comment. We have modified the wording of the relevant section as follows: “32.1% (8L: 16D), 32.3% (12L: 12D) and 32.8% (16L: 8D)”

Comments 6: Please check the sentence and match with the figure its contradictory.

Response 6: We thank the reviewer for this careful observation. We apologize for the error in the original manuscript where "larger" was incorrectly used instead of "lower". This has been corrected. The sentence now accurately reflects that the value was, in fact, greater than the others.

Comments 7: Better re-frame it as microbial composition.

Response 7: Thank you for your suggestion. We have revised the term "microbiological composition" to "microbial composition" in Figures 2 to 5. The modifications have been highlighted in the manuscript.

Comments 8: Better choose to re-frame it as significant difference.

Response 8: Thank you for your suggestion. We have revised the term "significant different" to "significant difference"

Comments 9: Line 235: addition of stop.

Response 9: Thank you for pointing this out. We have added a point at the end of the sentence.

Comments 10: The authors need to add the legends to the each segments because it cannot be understood what type of samples represented here.

Response 10: We appreciate your feedback. Figure 6 has been redone with an improved color scheme and an accompanying legend.

4. Response to Comments on the Quality of English Language

Point 1: Lines:207: Grammatical errors

Response 1: Thank you for pointing this out. We have corrected the grammatical error, and the change has been highlighted in the manuscript.

5. Additional clarifications

Finally, we sincerely appreciate your valuable comments. We have thoroughly revised the manuscript based on all suggestions. We sincerely apologize for the errors identified in the Materials and Methods section and have proactively communicated with the sequencing company to rectify these issues. We sincerely appreciate your keen attention to these details.

Reviewer 4 Report

Comments and Suggestions for Authors

This article investigates how different photoperiods and temperatures affect the antioxidant status and gut microbial community of the spotted mandarin fish (Siniperca scherzeri). The main contribution of this work is its demonstration of how different environmental conditions influence the physiological state of this important aquaculture species.

The article is well-structured and easy to read. The manuscript's strength lies in its comprehensive analysis of two factors: photoperiod and temperature. The authors' work has yielded compelling data on changes in the activity of various antioxidant enzymes and the composition of the intestinal microbiota. Such data are particularly relevant for cultivating aquacultured species, especially during critical developmental stages (e.g., larvae and fry), as they can help optimize cultivation technologies to minimize mortality and maximize productivity. The authors correctly point out that unfavorable environmental conditions for fish can lead to a compromised antioxidant status and the development of bacterial diseases, which can be caused by weakened immunity and an increase in opportunistic microorganisms within the host-associated microbial communities.

The work holds both scientific and practical value. However, to improve the manuscript's quality, the authors should clarify several factual points and expand the “Materials and Methods” section.

Specific comments:

- In the “Simple Summary,” the authors state that their results can serve as reference data for cultivating S. scherzeri (line 26). However, this claim is not developed further in the manuscript. To support this statement, the authors should expand the “Discussion” section to include brief recommendations for the photoperiod and temperature conditions that, based on their findings, are most favorable for S. scherzeri larvae.
- In the abstract (line 41) and discussion (lines 364–368), the authors state that an unfavorable photoperiod can lead to the proliferation of pathogenic bacteria (specifically, the genus Plesiomonas). The term “opportunistic bacteria” would be more accurate, as most microorganisms associated with fish diseases are natural inhabitants of the aquatic microbiome and mucous membranes that typically only become virulent when the host’s homeostasis is disrupted.
- For improved clarity, please add data on the initial and final size and weight of the S. scherzeri larvae to the “Materials and Methods” section (e.g., in subsection 2.1).
- To increase the practical value of the data, the authors should add information on the typical photoperiod and temperature conditions for this species. While some temperature data are mentioned in the “Discussion” (lines 325–338), consolidating this information earlier would provide a clearer justification for the chosen experimental parameters and allow readers to understand how they compare to the species' optimal range.
- The manuscript states that the experiment lasted 20 days. Please provide a clear justification for this duration, explaining why it is sufficient to obtain reliable results for the parameters studied.
- Please specify the number of fish used for the microbiome analysis.
- Please ensure all species names are italicized throughout the manuscript (e.g., lines 75, 77).

Author Response

1. Summary

Thank you very much for taking the time to review this manuscript. We would like to express our sincere gratitude to you for your time and insightful comments. Your valuable suggestions have greatly helped us in improving the quality of our manuscript.

2. Questions for General Evaluation

Reviewer’s Evaluation

Response and Revisions

Does the introduction provide sufficient background and include all relevant references?

Yes

Is the research design appropriate?

Yes

Are the methods adequately described?

Can be improved

We have revised the description of the Materials and Methods section to improve clarity.

Are the results clearly presented?

Yes

Are the conclusions supported by the results?

Yes

Are all figures and tables clear and well presented?

Yes

3. Point-by-point response to Comments and Suggestions for Authors

Comments 1: This article investigates how different photoperiods and temperatures affect the antioxidant status and gut microbial community of the spotted mandarin fish (Siniperca scherzeri). The main contribution of this work is its demonstration of how different environmental conditions influence the physiological state of this important aquaculture species.

The article is well-structured and easy to read. The manuscript's strength lies in its comprehensive analysis of two factors: photoperiod and temperature. The authors' work has yielded compelling data on changes in the activity of various antioxidant enzymes and the composition of the intestinal microbiota. Such data are particularly relevant for cultivating aquacultured species, especially during critical developmental stages (e.g., larvae and fry), as they can help optimize cultivation technologies to minimize mortality and maximize productivity. The authors correctly point out that unfavorable environmental conditions for fish can lead to a compromised antioxidant status and the development of bacterial diseases, which can be caused by weakened immunity and an increase in opportunistic microorganisms within the host-associated microbial communities.

The work holds both scientific and practical value. However, to improve the manuscript's quality, the authors should clarify several factual points and expand the “Materials and Methods” section.

Response 1: Thank you for your comment. We have meticulously revised the Materials and Methods section, including the sample processing and gut microbiota analysis procedures, to ensure precise alignment with the corresponding content in subsequent sections of the manuscript.

Comments 2: In the “Simple Summary,” the authors state that their results can serve as reference data for cultivating S. scherzeri (line 26). However, this claim is not developed further in the manuscript. To support this statement, the authors should expand the “Discussion” section to include brief recommendations for the photoperiod and temperature conditions that, based on their findings, are most favorable for S. scherzeri larvae.

Response 2: Thank you for this comment. We have carefully revised the Discussion section, and the modifications are marked in red in the updated manuscript.

Comments 3: In the abstract (line 41) and discussion (lines 364–368), the authors state that an unfavorable photoperiod can lead to the proliferation of pathogenic bacteria (specifically, the genus Plesiomonas). The term “opportunistic bacteria” would be more accurate, as most microorganisms associated with fish diseases are natural inhabitants of the aquatic microbiome and mucous membranes that typically only become virulent when the host’s homeostasis is disrupted.

Response 3: Thank you for your valuable feedback. The terminology accuracy you highlighted is indeed crucial. We fully agree that "opportunistic bacteria" is a more precise term than "pathogenic bacteria" in describing this ecological phenomenon. We have revised the term "pathogenic bacteria" to "opportunistic bacteria", with corresponding adjustments to more accurately reflect the ecological characteristics of these microorganisms.

Comments 4: For improved clarity, please add data on the initial and final size and weight of the S. scherzeri larvae to the “Materials and Methods” section (e.g., in subsection 2.1).

Response 4: Thank you for your comment. However, this particular parameter was not measured during our experiment. We would like to emphasize that all experimental fish were from the same batch, and were maintained under identical feeding conditions with strict control at every stage of the experimental process. We plan to conduct dedicated future studies focusing on the effects of photoperiod and temperature on growth of S. scherzeri.

Comments 5: To increase the practical value of the data, the authors should add information on the typical photoperiod and temperature conditions for this species. While some temperature data are mentioned in the “Discussion” (lines 325–338), consolidating this information earlier would provide a clearer justification for the chosen experimental parameters and allow readers to understand how they compare to the species' optimal range.

Response 5: We appreciate your valuable feedback. We have carefully discussed the points raised and have incorporated information on the typical photoperiod and temperature conditions into the manuscript.

Comments 6: The manuscript states that the experiment lasted 20 days. Please provide a clear justification for this duration, explaining why it is sufficient to obtain reliable results for the parameters studied.

Response 6: We would like to thank the reviewer for raising this question. The 20-day experimental period was determined primarily based on the following considerations: Firstly, according to previous studies and our pre-experimental results, a 20-day treatment duration is sufficient to induce stable and measurable significant effects of the photoperiod and temperature treatment on the oxidative stress levels and microbiota structure of the target species. Secondly, this period ensures that the experimental animals have adequate time to acclimate to the new environment (approximately 3-5 days), followed by exposure to a complete physiological response process.

Comments 7: Please specify the number of fish used for the microbiome analysis.

Response 7: Thank you for your comment. We have restructured the Materials and Methods section and added a new paragraph in subsection 2.1 to describe the specific procedures for sample processing and grouping.

Comments 8: Please ensure all species names are italicized throughout the manuscript (e.g., lines 75, 77).

Response 8: Thank you for your comment. We have italicized all species names in the third paragraph of the introduction and have carefully checked for other occurrences of species names throughout the manuscript.

4. Response to Comments on the Quality of English Language

Point 1: None

Response 1: None

5. Additional clarifications

We sincerely thank you for this valuable feedback. Regarding the point on sample size, we sincerely apologize that we are unable to address it directly within the revision. Conducting the necessary supplemental experiments is contingent upon specific seasonal conditions, which will not be available until next summer. Therefore, we regret that we cannot add more samples or perform additional analyses at this time. We fully acknowledge that this limitation may affect the interpretation of our findings, and we will give paramount importance to ensuring adequate sample size in our future research plans. Once again, we are truly grateful for your constructive suggestion, which has been crucial in identifying this key aspect for improvement.

Round 2

Reviewer 1 Report

Comments and Suggestions for Authors

Thank you for taking the time to incorporate the suggestions from the previous review. This revised version communicates the work more clearly and reads with better flow and structure. The topic is timely and globally relevant, especially considering the current challenges in food production. I truly appreciate the effort reflected in the manuscript, and I believe it meets the journal’s standards for publication.

Reviewer 2 Report

Comments and Suggestions for Authors

Dear Author,

The current manuscript is well-designed. However, it had some shortcomings. It could have been supplemented with some analysis. Still, it's quite good as is. I'm happy to accept your manuscript.

Best regards.

Reviewer 3 Report

Comments and Suggestions for Authors

I recommend acceptance of this manuscript.